# Thermal Field Reconstruction on Microcontrollers: A Physics-Informed Digital Twin Using Laplace Equation and Real-Time Sensor Data

**DOI:** 10.3390/s25165130

**Published:** 2025-08-19

**Authors:** Victor H. Benitez, Jesus Pacheco, Agustín Brau

**Affiliations:** Department of Industrial Engineering, Universidad de Sonora, Hermosillo 83000, Mexico; jesus.pacheco@unison.mx (J.P.); agustin.brau@unison.mx (A.B.)

**Keywords:** physics-informed digital twin, thermal field estimation, embedded systems, finite difference method, real-time sensing

## Abstract

**Highlights:**

**What are the main findings?**
The embedded system accurately reconstructs internal temperature distributions from sparse boundary sensor data in real time.The proposed architecture achieves stable and low-latency thermal simulation using physics-based modeling on resource-constrained microcontrollers.

**What is the implication of the main finding?**
Enables the deployment of physics-based digital twins in low-power edge devices without reliance on cloud infrastructure.Facilitates affordable and interpretable thermal diagnostics and monitoring in educational, prototyping, and industrial environments.

**Abstract:**

This paper presents a physics-informed digital twin designed for real-time thermal monitoring and visualization of a metallic plate. The system comprises a physical layer consisting of an aluminum plate equipped with thermistors to capture boundary conditions, a computational layer that implements the steady-state Laplace equation using the finite difference method, and an embedded execution framework deployed on a microcontroller that utilizes Direct Memory Access-driven ADC for efficient concurrent acquisition. The computed thermal field is transmitted through a serial interface and displayed in real time using a Python-based visualization interface. The Steinhart–Hart model was used to experimentally characterize the sensors, ensuring accuracy in the boundary condition acquisition. While the current formulation is restricted to steady-state conditions, it enables accurate spatial reconstructions with acceptable error margins and demonstrates operational concurrency with the physical system. The compact and modular architecture allows adaptation to other physical domains governed by elliptic PDEs, making it suitable for educational applications, diagnostic prototyping, and embedded edge deployments.

## 1. Introduction

The two-dimensional Laplace equation is fundamental for simulating steady-state physical phenomena such as heat conduction, electrostatics, and incompressible fluid flow. Numerical solutions of this equation yield spatial temperature distributions that are vital for understanding and managing thermal behavior in engineered systems. Until a few years ago, solving this equation in real-time required powerful desktop computers or workstations. However, recent advancements in embedded computing have fostered the development of innovative strategies for executing these solvers on low-power, cost-effective microcontrollers.

This study presents an innovative embedded implementation of a physics-informed digital twin, designed to monitor and visualize the temperature distribution across a metallic plate governed by the two-dimensional Laplace equation. Unlike conventional data-driven digital twins, which often rely on black-box machine learning models and extensive datasets, this implementation is based on first-principles modeling (a physically informed system).

The physical layer consists of a one mm-thick aluminum plate equipped with thermistors distributed along its boundaries. The thermistors are experimentally calibrated using the nonlinear Steinhart–Hart model, which requires the experimental measurement of three known resistance-temperature pairs to solve a system of equations and obtain a precise conversion from resistance to temperature.

A Python-based graphical interface communicates with the microcontroller via serial communication to visualize the results. The interface renders the computed temperature distribution as a dynamic thermal map, enabling intuitive interpretation and real-time monitoring.

This architecture enables physics-informed digital twins in constrained environments, with potential applications in education, rapid prototyping of thermal systems, and real-time monitoring of photovoltaic modules, as well as thermal testing of construction materials and embedded thermal diagnostics. The system eliminates the need for external computation and significantly reduces data transmission overhead by sending only the computed temperature matrix, rather than raw sensor data, thereby bringing physical modeling directly to the edge. The proposed approach enhances scalability, minimizes latency, and enables the efficient and autonomous operation of the digital twin in embedded contexts. The system exemplifies how embedded digital twins can reconstruct spatial fields from sparse sensor data and can be adapted to other domains involving steady-state partial differential equations (PDEs).

The proposed implementation represents a foundational layer of a physics-informed digital twin, focusing on real-time state concurrency, embedded physical modeling, and closed-loop data processing. Although data persistence and cognitive adaptation are not yet included, the system satisfies key operational criteria for embedded DTs and provides a modular architecture for future development extensions.

The remainder of the paper is structured as follows: Section 2 reviews related work and the state of the art. Section 3 introduces the theoretical model and system architecture. Section 4 describes the experimental implementation, including sensor calibration, hardware/software configuration, and visualization strategy. Section 5 discusses implications, limitations, and future directions. Finally, Section 6 presents the conclusions.

## 2. Related Work

The concept of digital twins (DTs) has gained significant traction in recent years, in the context of thermal system monitoring and control. References, such as [1,2,3,4], can be found where several approaches are presented, all of which rely on high computing resources. DTs are virtual replicas of physical systems that enable real-time monitoring, simulation, and optimization. They are broadly categorized into data-driven, physics-based, and hybrid approaches [5].

### 2.1. Data-Driven Digital Twins

Data-driven digital twins depend on machine learning algorithms trained on historical and real-time data to predict system behavior. For example, artificial neural networks (ANNs) and recurrent neural networks (RNNs) have been utilized to forecast thermal load profiles in buildings and industrial systems [6]. Support vector machines (SVMs) and decision trees are utilized in digital twin frameworks to identify anomalies in HVAC systems and enhance energy consumption efficiency [7]. These applications demonstrate how data-driven approaches can identify intricate thermal behavior patterns, providing real-time monitoring and control solutions.

### 2.2. Physics-Based Digital Twins

Physics-based digital twins (PBDTs) employ mathematical models rooted in fundamental physical principles to simulate system behavior. These models frequently require solving partial differential equations (PDEs) that represent the underlying physical phenomena. For example, the Laplace equation has been applied in robotics for path planning, improving its properties to model potential fields that guide robots through environments while avoiding obstacles [8]. In industrial settings, PBDTs are used to simulate and manage heating processes. By integrating the principles of heat transfer, such as conduction and radiation, these DTs can model temperature distributions within heating systems like furnaces. This allows for precise control of the heating process, enhancing both efficiency and product quality. For example, a study developed a physics-informed DT for industrial heating processes, integrating physical models with machine learning techniques to improve temperature prediction accuracy [9].

### 2.3. Hybrid Approaches for Digital Twins

Hybrid digital twins combine data-driven and physics-based modeling methods to power their strengths. Although physics-based models ensure interpretability and comply with fundamental principles, data-driven models demonstrate adaptability and can manage complex datasets. Combining these approaches facilitates enhanced prediction accuracy, robustness, and generalization, particularly in dynamic environments where either approach alone may be insufficient.

Recent studies have explored this synergy in the context of thermal systems. For example, a hybrid digital twin framework was developed to model and control transient thermal systems, where deep neural networks were trained on stationary data to estimate heat transfer coefficients in real time. These coefficients were then used within a finite element (FE) physical model to regulate maximum temperatures under dynamic heat fluxes, ensuring operational safety and thermal efficiency in 1D, 2D, and 3D domains [10]. In aerospace applications, a hybrid digital twin architecture is proposed to enhance the prediction of satellite temperature fields. This approach merges data-driven models with physical field simulations, ensuring high fidelity across both spatial and temporal dimensions [11].

Recent technical literature and industry discussions indicate that developing hybrid DTs improves modeling accuracy and decision-making in complex systems. These systems face considerable challenges due to uncertain boundary conditions, time-varying dynamics, and nonlinearities [5].

### 2.4. Numerical Solutions of PDEs on Embedded Systems

Solving Laplace’s equation on embedded systems is challenging due to the intensive computation and memory required for numerical solutions. Traditionally, powerful processors handle such PDEs, but recent advances have enabled embedded solutions. For instance, researchers have developed custom low-power accelerators that directly solve discretized Laplace/Poisson equations in silicon [12]. On the other hand, the authors of the reference [13] have explored analog computing approaches. They developed an analog finite-difference time-domain solver chip for a nonlinear shock-tube PDE, with a microcontroller used to calibrate the system. These researchers demonstrate advanced PDE solvers on embedded platforms, utilizing digital and analog hardware acceleration to manage large linear systems from Laplace’s equation.

Even low-cost microcontrollers, though with coarse resolution, can solve Laplace’s equation (or similar PDEs) in real time. Simple, memory-local iterative methods like Jacobi or Gauss–Seidel relaxation are implementable on microcontroller units (MCUs) to approximate Laplace’s solution on a grid. For instance, an Arduino demo solved a 2D fluid flow equation using a finite difference scheme on a 25 × 25 grid [14]. Although performance is limited, integrating a PDE solver into a microcontroller enables autonomous sensor and actuator devices that can, for example, compute temperature or potential fields on-device without relying on cloud or PC resources.

### 2.5. Finite Difference Methods on Microcontrollers

Among numerical techniques for PDEs, the finite difference method (FDM) is particularly suitable for implementation on microcontrollers due to its straightforward and iterative nature. FDM replaces the continuous Laplace operator ∇^2^u = 0 with discrete differences on a mesh, resulting in linear equations that relate each point to its neighbors [12]. FDM stencils, such as the five-point stencil in 2D Laplace problems, require each grid node update to be a simple arithmetic average or weighted sum of neighbor values—operations manageable by a modest 32-bit MCU. Implementing an FDM solver involves using an iterative solver (like Jacobi or Gauss–Seidel) to relax the solution across the grid. By applying local updates, these algorithms can halt when the change in the solution is below a specified threshold, ensuring that they conform to the limited throughput of microcontrollers.

Over the past ten years, academic and industrial research has demonstrated that microcontrollers and embedded processors can perform FDM computations for PDEs in real-time, given specific constraints. On the more advanced side, custom hardware accelerators combine numerous simple processing elements to update grid points simultaneously [15]. In reference [15], Mu and Kim presented a reconfigurable-bit-width 21 × 21 processing-element lattice that maps directly onto a 2D grid for FDM computations. Each tiny processing element handles the update of one cell of the discretized domain and communicates with its four neighbors, enabling massive parallelism for iterative methods.

### 2.6. Computing Thermal Boundary Conditions

It is necessary to apply thermal boundary conditions to address heat distribution challenges, particularly in embedded systems where Laplace’s equation serves as a specific steady-state model. Thermistors are a popular choice for temperature sensors at boundaries due to their small size and high sensitivity. In embedded PDE applications, thermistor measurements can be utilized to set Dirichlet boundary values for fixed temperatures or Neumann conditions at the edges of a simulated domain [16]. However, raw thermistor signals are highly nonlinear with respect to temperature due to the semiconductor nature of the material from which the thermistor is made. The Steinhart–Hart method is a widely used empirical model for converting thermistor resistance into an accurate temperature reading [17]. This model expresses the temperature as a cubic polynomial as(1)$$\frac{1}{T}=A+B\ln(R)+C{\left(\ln(R)\right)}^{3}$$
with A, B, and C as sensor-specific coefficients; T refers to the temperature in Kelvin; and R denotes the electrical resistance.

## 3. Theoretical Background

### 3.1. Laplace’s Equation in Heat Transfer

Laplace’s equation is a second-order linear partial differential equation (PDE) that describes the behavior of scalar fields in the absence of sources or sinks. In two dimensions, it takes the form(2)$$\frac{\partial^{2}T}{\partial x^{2}}+\frac{\partial^{2}T}{\partial y^{2}}=0$$

This formulation is based on Fourier’s law and the conservation of energy, assuming the system has reached thermal equilibrium. In the context of digital twins, this PDE serves as the physics-informed backbone for estimating the internal temperature field from limited boundary measurements.

To implement this numerically on embedded hardware, the domain (a 2D metal plate) is discretized into a grid. Using the finite difference method, the second-order partial derivatives are approximated as:(3)$$\begin{array}{l} \frac{\partial^{2}T}{\partial x^{2}}\approx \frac{T_{i+1,j}-2T_{i,j}+T_{i-1,j}}{\Delta x^{2}}\\ \frac{\partial^{2}T}{\partial y^{2}}\approx \frac{T_{i,j+1}-2T_{i,j}+T_{i,j-1}}{\Delta y^{2}} \end{array}$$

Assuming a square mesh, Δx=Δy, and combining both approximations, the discrete Laplace operator is obtained as(4)$$T_{i+1,j}+T_{i-1,j}+T_{i,j+1}+T_{i,j-1}-4T_{i,j}=0$$

This algebraic expression is valid for all interior nodes. Boundary conditions are imposed using real-time sensor data from thermistors distributed along the edges of the plate. These readings serve as Dirichlet boundary conditions.

### 3.2. Embedded Implementation and Convergence

The discrete system of equations is solved iteratively using Liebmann’s method. To compute the update rule for interior nodes, Equation (5) is applied(5)$$T_{i,j}^{\left(n+1\right)}=\frac{1}{4}\left(T_{i+1,j}^{\left(n\right)}+T_{i-1,j}^{\left(n\right)}+T_{i,j+1}^{\left(n\right)}+T_{i,j-1}^{\left(n\right)}\right)$$

The $\left(n+1\right)$ term is solved iteratively for j=1 to n and i=1 to m. Since Equation (4) is diagonally dominant, this procedure will ultimately converge on a stable solution. To enhance the convergence rate, overrelaxation is employed by applying (6) after each iteration.(6)$$T_{i,j}^{\left(n\right)}=\lambda T_{i,j}^{\left(n\right)}+(1-\lambda )T_{i,j}^{\left(n-1\right)}$$
where λ is a weighting factor that is set between 1 and 2. Convergence is determined when the maximum absolute difference between iterations falls below a tolerance criterion(7)$$\left|{\left(\varepsilon_{tol}\right)}_{i,j}\right|=(100)\left|\frac{T_{i,j}^{\left(n\right)}-T_{i,j}^{\left(n-1\right)}}{T_{i,j}^{\left(n\right)}}\right|$$

This tolerance, combined with uniform grid spacing and proper floating-point management, ensures numerical stability even on resource-constrained microcontrollers. The physics-informed nature of the model ensures interpretability, physical consistency, and robustness in reconstructing internal states from limited boundary measurements, qualifying the system as a true digital twin.

### 3.3. Embedded Systems Architecture for Real-Time Thermal Monitoring

Implementing DTs in embedded systems presents several challenges due to limitations in computational resources, memory availability, and power consumption. However, recent advancements in microcontroller design have significantly enhanced the feasibility of deploying real-time monitoring and control solutions at the edge.

The STM32 microcontroller series, based on the ARM Cortex-M architecture, spans a wide range of performance classes. From the ultra-low-power Cortex-M0/M0 + (STM32F0/L0 series), suitable for simple sensing and control, to the high-performance Cortex-M7 cores (e.g., STM32F7, STM32H7), capable of DSP-level operations and floating-point arithmetic, the STM32 ecosystem offers a scalable platform for embedded implementations. Mid-range variants, such as the Cortex-M3/M4 (STM32F1/F3/F4 series), offer a balance between computational capability, peripheral variety, and energy efficiency [18]. Additionally, STM32 technology includes advanced features like dual ADCs, Direct Memory Access (DMA), floating-point units (FPU), and digital signal processing (DSP) instructions embedded in a single core, which are critical for real-time signal acquisition and numerical computations [19].

Compared to other providers, such as Microchip’s PIC microcontrollers, STM32 MCUs offer higher clock speeds, broader 32-bit architecture support, and more advanced peripherals. While SoCs (e.g., Raspberry Pi) and dedicated DSPs (e.g., TI C6000 series) can provide greater processing capabilities, they often exceed power, cost, and complexity constraints for embedded applications. STM32 MCUs bridge this gap by offering DSP-like performance in a low-power, suitable for field-deployable systems [20,21].

For instance, the STM32F3XX microcontroller family, equipped with dual high-speed ADCs and integrated DMA controllers, has been successfully employed in thermal applications requiring synchronized multichannel sampling. Coupling such hardware capabilities with numerical methods like the finite difference method enables the implementation of compact, low-latency embedded systems that simulate physical heat transfer phenomena in real-time [22].

## 4. Methodology

The methodology is constituted by four key components: (i) physical system and boundary sensing, (ii) embedded numerical solver, (iii) communication and data exchange, and (iv) visualization interface. The core computational framework is embedded in a low-power microcontroller that integrates sensor acquisition, finite-difference numerical modeling, and a real-time thermal block.

### 4.1. Experimental Embedded Setup for Real-Time Thermal Monitoring

The physical system consists of a planar metallic surface acting as a 2D conductive domain governed by Laplace’s equation under steady-state conditions(8)$$\Delta^{2}T\left(x,y\right)=0$$

To emulate physical boundary conditions, sensors are positioned along the edges of the medium to monitor fixed temperature. The thermistor sensors are of the NTC type and are characterized using the Steinhart–Hart model, as shown in Equation (1). These sensors connect to the analog inputs of the microcontroller through precision voltage dividers. Each reading is sampled via dual 12-bit ADCs operating in DMA mode on the STM32F303, enabling efficient and parallel acquisition of temperature boundary values.

### 4.2. Embedded Numerical Solver

The thermal field inside the conductive medium is computed using a finite difference method (FDM), where the Laplacian operator is discretized over a uniform grid using Equation (5).

The domain is modeled as a 2D array of nodes, with each node representing a discretized temperature point. Boundary values are updated dynamically based on thermistor readings, while interior nodes are iteratively updated until the steady-state convergence criteria are met. The iterative solver is optimized for low memory usage and computational latency, employing in-place array updates and integer arithmetic where feasible. Convergence thresholds are pre-calibrated to strike a balance between resolution and computational load.

The real-time performance of the proposed digital twin is enabled by the internal architecture of the STM32F303RET6 microcontroller, which integrates a high-speed Arm Cortex-M4 core with an advanced bus matrix and dual DMA controllers. Figure 1 illustrates a multi-layer AHB bus matrix that interconnects peripherals, memory blocks, and system buses, enabling high-throughput, low-latency communication throughout the system.

A key architectural aspect of this implementation is the presence of two analog-to-digital converters (ADC1 and ADC2), both of which can function in independent scan mode. These ADCs are linked to DMA1 and DMA2 controllers, respectively, enabling simultaneous data collection from various sensors without CPU intervention.

The system ensures continuous, non-blocking acquisition of boundary temperature data by configuring each ADC to operate in direct DMA mode. The DMA engines autonomously transfer data from the ADC data registers to memory buffers, freeing the CPU to execute the finite-difference Laplace solver concurrently. This architectural parallelism minimizes system latency and is critical for the physical consistency of the digital twin.

### 4.3. Communication and Data Exchange

The estimated thermal field is transmitted via a UART interface for real-time monitoring. The STM32F303 MCU employs a communication protocol to serialize the two-dimensional temperature matrix into byte streams, thereby ensuring synchronization and fault-tolerant transmission. For the purposes of debugging and calibration, the microcontroller also records boundary sensor values and solver convergence metrics at fixed intervals. Additionally, this interface supports optional I^2^C expansion for configurations involving multiple sensors.

The communication phase of the digital twin architecture methodology is based on the efficient coupling between the embedded system and the external digital representation. After reconstructing the internal thermal field from boundary data using the embedded Laplace solver, the result must be transferred to a host system for visualization and further analysis. This stage closes the feedback loop between the physical and digital layers, preserving the synchronicity of the digital twin.

The communication protocol utilizes a low-latency transmission scheme, employing a serial interface to transmit the computed temperature matrix. In formal terms, let T(x,y) be the continuous solution that is approximated numerically as a discrete grid Ti,j. Once convergence of the numerical scheme is reached, the matrix Ti,j is linearly serialized into a vector form using a linear transformation given by Equation (9)(9)Tvec=vec(Ti,j)∈ℝN2
where N2 indicates a square grid of sensors. This vector contains the temperature values of the interior mesh nodes (excluding boundary points, which are physically measured) and is transmitted over a reliable and lightweight point-to-point protocol to the host system.

The design prioritizes minimal communication overhead, aligning with the embedded constraints of limited buffer sizes and energy efficiency. No external memory or higher-layer communication stacks (e.g., TCP/IP) are employed, ensuring that data transfer latency remains bounded and predictable.

On the host side, a software interface reconstructs the 2D thermal field from the received vector using the inverse mapping(10)$$T_{i,j}={\mathrm{vec}}^{-1}(T_{\mathrm{vec}})$$

Equation (10) implements a reconstruction vector that feeds into a visualization layer, which interprets the temperature distribution as a color-coded heatmap, simulating a virtual thermal camera. Visualization is a fundamental element of the digital twin architecture, providing real-time insights into the evolution of the physical system’s state based on sensed data.

From a methodological perspective, the communication layer provides the observability of the physical system, translating raw sensor data and internal model computations into an intelligible spatial representation. Moreover, since the model relies on physical laws rather than empirical mappings, the resulting visualization is grounded in physically consistent behavior, enhancing its interpretability.

This layer completes the digital twin cycle by creating a two-way channel: the embedded system communicates with the digital interface, and visual feedback can influence operational or experimental decisions in the physical environment. The proposed architectural modularity also allows for future enhancements, such as feedback-based control, anomaly detection, and integration into broader cyber-physical systems.

Figure 2 displays the methodology employed in this study.

## 5. Results

### 5.1. System Setup and Execution

The entire system was installed on a custom fixture that supports a 1 mm aluminum plate equipped with 12 thermistors (three on each edge). These sensors measure real-time temperatures and are used as Dirichlet boundary conditions for the numerical solver implemented within the embedded system. The spatial domain of the plate was discretized into a uniform N × N grid, with N = 5, resulting in 25 nodes. Figure 3 illustrates the thermal field reconstruction scenario: subfigure (a) presents the schematic representation of the 5 × 5 grid used in the numerical model, including estimated interior values (red dots) and sensor-measured boundary values (circled); subfigure (b) shows the physical implementation of the sensor array on the metallic plate.

Figure 3a shows the estimated temperature (red dots) calculated by the algorithm, along with the actual temperatures (circled values) measured by the thermistors. This is represented as a matrix, with the plate divided into sections as$$\left[\begin{array}{ccccc} T_{1} & T_{2} & T_{3} & T_{4} & T_{5}\\ T_{6} & T_{7} & T_{8} & T_{9} & T_{10}\\ T_{11} & T_{12} & T_{13} & T_{14} & T_{15}\\ T_{16} & T_{17} & T_{18} & T_{19} & T_{20}\\ T_{21} & T_{22} & T_{23} & T_{24} & T_{25} \end{array}\right]$$

Notice that $T_{13}$ is the inner estimated temperature. The resolution (grid size) was selected to balance the computational load of the embedded processor with the spatial resolution needed to capture meaningful gradients.

In the implemented configuration, 12 thermistors were distributed along the boundary of the metallic plate to provide Dirichlet conditions for the numerical solver. These sensors were placed at specific grid positions: $T_{2},\;T_{3},T_{4}$ (top edge), $T_{10},\;T_{15},T_{20}$ (right edge), $T_{22},\;T_{23},T_{24}$ (bottom edge), and $T_{6},\;T_{11},T_{16}$ (left edge). The microcontroller calculated the remaining 13 nodes using the finite difference method. This arrangement yields a 5 × 5 uniform mesh, where the spatial spacing ∆x=∆y corresponds to four equal divisions along each axis of the square domain.

The decision to employ a 5 × 5 grid was motivated by the trade-off between spatial resolution and computational feasibility on a resource-constrained embedded platform. The Liebmann solver used for estimating internal values has a per-iteration complexity of $O\left(N^{2}\right)$, and finer meshes would increase memory usage and convergence time quadratically. While higher-resolution grids can be achieved by adding more sensors along the boundary, the chosen configuration guarantees convergence within real-time constraints and reduces system complexity.

Sensors were linked to the ADC1 and ADC2 peripherals of the microcontroller, which were configured for DMA-based sampling to avoid any interruptions in the numerical computation loop. The grid size was determined based on the available RAM and processing power of the MCU. Each node of the internal domain was updated using a Jacobi method, and the maximum difference between iterations was monitored to evaluate convergence.

The embedded system implements the Liebmann method, a variant of the Jacobi iterative solver, to compute the steady-state temperature distribution. At each iteration, every interior node $T_{i,j}$ is updated by averaging the temperature values of its four immediate neighbors according to Equation (5).

To accelerate convergence, a successive over-relaxation technique is applied as described by Equations (6) and (7). The iterative process continues until the maximum change between iterations falls below a predefined threshold $\varepsilon_{tol}$, ensuring numerical stability.

Once convergence is achieved, the temperature matrix is flattened and transmitted over UART to a Python-based visualization layer, enabling real-time rendering of the heat distribution. The system performs acquisition, computation, and communication autonomously, without external processing.

### 5.2. Boundary Condition Measurement

Boundary conditions play a critical role in ensuring the accuracy of the numerical model implementation. Each of the twelve thermistors attached to the edges of the metallic plate was experimentally characterized, represented by Equation (1). The resistance $R_{t}$ of each sensor is calculated from the ADC value using a voltage divider with a known series resistor, and temperature is estimated using Equation (1). The unknowns A, B, and C are the Steinhart–Hart coefficients that uniquely characterize each thermistor. These coefficients must be obtained through experimental calibration by measuring the resistance of the thermistor at three known temperatures.

Let the three resistance R and temperature T calibration points be $\left(T_{1},R_{1}\right)$, $\left(T_{2},R_{2}\right)$, and $\left(T_{3},R_{3}\right)$. For each point, Equation (11) is applied. This results in a system of three nonlinear equations, which can be expressed in matrix form as(11)$$\left[\begin{array}{ccc} 1 & \ln(R_{1}) & \ln^{3}(R_{1})\\ 1 & \ln(R_{2}) & \ln^{3}(R_{2})\\ 1 & \ln(R_{3}) & \ln^{3}(R_{3}) \end{array}\right]\left[\begin{array}{c} A\\ B\\ C \end{array}\right]=\left[\begin{array}{c} \frac{1}{T_{1}}\\ \frac{1}{T_{2}}\\ \frac{1}{T_{3}} \end{array}\right]$$

Solving this system yields coefficients A, B and C, enabling the embedded system to estimate real-time temperatures from ADC computations accurately. In Figure 4, an example of sensor calibration is displayed with the coefficients computed.

### 5.3. Real-Time Temperature Field Computation

This phase entails running the numerical model integrated within the microcontroller to calculate the steady-state temperature distribution over the metallic plate. The previously outlined architecture enables the effective collection of boundary values using dual ADCs in DMA mode, which supply the real-time simulation engine.

The temperature computation relies on a finite-difference solution to the two-dimensional Laplace equation, whose physical model is determined by Equation (1). The spatial domain of the plate is discretized into a 5 × 5 grid. Boundary conditions at the outermost points are updated with thermistor values, while internal nodes are updated iteratively using Liebmann’s method while computing Equations (7)–(9). It is worth noting that λ = 1.5 was experimentally determined to balance convergence speed and numerical stability.

Figure 5 illustrates the experimental configuration employed during the initial real-time assessments. Figure 5a,b depict a close-up of the aluminum plate, which was equipped with calibrated thermistors distributed at the outer edges, secured with thermal tape, exposed to external heat sources using alcohol lamps, and wired to dual ADC channels configured in DMA mode. Figure 5c shows a thermal image captured by a camera to assess the temperature gradient as an initial estimate and support the early insights of the experimental setup (one lamp located at the bottom edge). Figure 5d shows the gradient image generated by the digital twin, with one lamp positioned at the left edge and another at the bottom edge. These sensors captured the boundary conditions in real time, while the interior temperature values were calculated iteratively on the microcontroller. A dedicated thermistor, not used by the solver, was placed in the center of the plate to act as a reference or witness sensor. This element enabled direct comparison between the computed and actual temperatures at the core of the domain.

After convergence was achieved, the data were visualized using a custom Python graphical interface that receives the reconstructed temperature vector from the microcontroller. Figure 5 shows the resulting thermal field, highlighting the interpolated internal temperatures and comparing the actual reference value (31.19 °C) with the estimated center temperature (29.43 °C), which results in an observed error of +1.76 °C. These initial results confirm that the embedded solver can reliably approximate the spatial thermal gradient using only boundary data, validating the methodology for further testing and adaptation of domain-specific applications.

### 5.4. Embedded Schedule Logic

Once both ADCs complete their DMA transfers, flags are raised, triggering the reconstruction of boundary temperature values from the acquired voltages using the Steinhart–Hart model. These values are mapped onto the external edges of the discretized domain. The interior of the grid is then updated according to the numerical solver logic.

This real-time loop is executed within the main application thread and consists of the following logical steps:DMA Completion Detection, where the embedded system waits for both ADC1 and ADC2 to complete DMA transfers.Boundary Assignment, where raw ADC values are converted into temperatures and assigned to border positions.Finite Difference Iteration, where the temperature of internal nodes is updated via the iterative solver.UART Transmission, where the resulting 5 × 5 temperature grid is serialized and transmitted to a host interface.DMA Restart, where the ADC readings are re-triggered to maintain a continuous acquisition cycle.

The previously mentioned logic sequence steps are expressed in Algorithm 1 as follows.
**Algorithm 1.** Embedded Schedule LogicWHILE True: IF adc1_done AND adc2_done:  boundary_values = convert_adc_to_temp(adc1_data, adc2_data)   assign_boundary_conditions(grid, boundary_values)   FOR *i* = **1** to L-**2**:     FOR *j* = **1** to L-**2**:      *T*_new = average_neighbor_temperatures(grid, *i*, *j*)      *T*_relaxed = lambda × *T*_new + (**1**-lambda) × *T*_old[*i*][*j*]      grid[*i*][*j*] = *T*_relaxed   send_grid_via_UART(grid)   restart_ADC_DMA()

### 5.5. Real-Time Thermal Visualization Interface

#### 5.5.1. Symmetrical Perturbations Experiment

To enable real-time visual feedback of the computed temperature field, a host-side visualization interface was implemented using Python and the Matplotlib v3.9.2 library, using an interface that communicates with the embedded system via UART over a serial connection. The embedded system transmits a linear array of 25 temperature values, corresponding to a 5 × 5 mesh representation of the metallic plate, at each iteration. This resolution can be modified depending on the number of boundary sensors used and the resolution required to represent the full internal temperature field estimated by the microcontroller.

Upon receiving a complete set of values, the Python program reshapes the flat vector into a 2D array, applies a light Gaussian filter for spatial smoothing, and then updates the heatmap in real time. Each cell in the mesh displays the temperature value with a white label overlay, providing both qualitative and quantitative insight into the gradient evolution. In the current configuration, the system generates one thermal image every 0.5 s, based on a total of 1200 samples collected during a 10 min experiment. This processing time includes boundary data acquisition via dual ADCs, iterative numerical solving with the Jacobi method and over-relaxation, convergence checking, and UART transmission. The update rate can be modified by adjusting the convergence tolerance, grid resolution, or data transmission settings. Algorithm 2 executes the logic behind the visualization interface.
**Algorithm 2.** Real-Time Thermal Visualization InterfaceInitialize serial port **and 5 × 5** temperature matrixInitialize interactive heatmap using matplotlibLoop indefinitely:  Wait **for 25** valid float values **from serial input**  If complete **and** valid data received:    Reshape into **5 × 5** matrix    Apply Gaussian smoothing    Update color map **and** text labels on plot

To assess the fidelity of the internal temperature estimation computed by the embedded system, a temporal analysis was conducted by comparing the simulated temperature at the center of the plate ($T_{13}$) against a calibrated reference sensor placed at the same location. In the experimental setup, four flame sources were applied symmetrically at the edges of the metallic plate for 10 min, establishing boundary conditions that propagated inward. The evolution of $T_{13}$ over time was plotted alongside the reference temperature, revealing that the embedded solver effectively captured the general heating trend and dynamic response of the plate (Figure 6a). However, discrepancies emerged after the initial transient phase, with the absolute error stabilizing around 3–4 °C once the temperature field reached a steady state. The real-time error was also calculated and visualized, providing a quantitative measure to evaluate the accuracy of the internal node approximation compared to the physical sensor (Figure 7).

Further insight is shown in Figure 6c,d by the reconstructed thermal gradient visualizations generated by the digital twin system at three representative time points. After 31.5 s from the start of the experiment (Figure 6a,c, sample 63), the plate exhibited relatively low and homogeneous temperatures, with the estimated center temperature (26.71 °C) closely matching the reference value (27.46 °C), resulting in a minimal absolute error of 0.75 °C (Figure 7). As the heat propagated from the boundaries (sample 660), the internal field developed a stronger gradient, and the estimated center value (56.06 °C) began to deviate from the measured 60.82 °C, yielding a larger error discrepancy of 4.76 °C (Figure 7). By the final stage (sample 1190), the system approached thermal equilibrium, and although the center temperature estimate improved to 58.73 °C, a residual error of 3.58 °C persisted with respect to the reference (62.31 °C) as displayed by Figure 6a,d and validated by the error performance behavior.

To further understand the dynamics of thermal conduction across the plate, a focused analysis was conducted along a linear path from the top edge $\left(T_{2}\right)$ toward the center $\left(T_{13}\right)$, passing through intermediate points $\left(T_{7},T_{12}\right)$. The temporal evolution of these points revealed the progressive inward diffusion of heat, where each subsequent point exhibited a delayed rise in temperature corresponding to its distance from the boundary. This edge-to-center propagation curve not only validates the expected physical behavior dictated by Fourier’s law but also serves as a diagnostic trajectory to assess how well the model captures the transient performance of thermal conduction (Figure 6b). The embedded system successfully mirrored this progression, though with minor temporal offsets and underestimations at intermediate points, highlighting the limitations of coarse spatial discretization. Nevertheless, the shape and sequence of thermal response closely matched expectations, reinforcing the validity of the model.

It is important to note that the numerical solver implemented in the embedded system is based on Laplace’s equation in two dimensions, as expressed in Equation (8). This formulation assumes steady-state heat conduction, where no internal heat generation or time-dependent changes are present. As such, the model inherently neglects transient thermal behaviors during the early stages of heating and converges toward an equilibrium solution as the system stabilizes. This theoretical foundation explains the high fidelity of the spatial distribution at later times, but also the initial discrepancies observed when the plate is still undergoing dynamic thermal transitions. The fidelity of the embedded approximation thus depends not only on spatial resolution and boundary accuracy, but also on the validity of the steady-state assumption over the timescales of interest.

Moreover, the experiment was deliberately conducted in a real laboratory environment subject to uncontrolled disturbances, such as external airflow from opened doors and exposure to air conditioning currents. These perturbations were intentionally preserved during the trials to evaluate the model’s robustness under non-ideal conditions, as typically encountered in industrial environments. While more accurate results could be obtained under strictly controlled thermal conditions, the current findings highlight the resilience of the embedded model in the presence of realistic environmental noise. This characteristic is essential for practical deployment, where such fluctuations are unavoidable. The ability to accurately reflect thermal propagation and gradient formation under such disturbances further demonstrates the robustness and applicability of the proposed system in real-world settings. The thermal gradients corresponding to the beginning and the end of the 10 min test are displayed by the digital twin and are shown in Figure 6c,d.

To improve the quantitative assessment of the reconstructed thermal field, a detailed error analysis was performed using the embedded thermistor located at the center of the metallic plate (node $T_{13}$) shown in Figure 7, which acts as a witness sensor. This point is the farthest from the boundary sensors and therefore poses the greatest challenge for the numerical solver. Figure 7 features four subplots: (a) the time-resolved error between the reconstructed and reference temperature at $T_{13}$; (b) the magnitude of the absolute error; (c) a baseline plot showing the Root Mean Square Error (RMSE); and (d) the percentage error over time. The metrics calculated during the 10 min experiment are as follows: RMSE: 3.57 °C, MAE: 3.35 °C, and MPE: 5.97%. These metrics confirm that the embedded solver maintains a reasonable level of accuracy in real-world conditions, supporting the proposed approach as a practical tool for real-time field reconstruction in environments with limited resources.

Figure 8 presents the output of the digital twin graphic interface as a series of thermal snapshots captured during the first three minutes of the experiment under symmetric heating conditions. In this configuration, four alcohol flame sources were placed at each edge of the metallic plate, simulating an evenly distributed thermal excitation.

Each frame corresponds to a specific time mark: 30 s (sample 60), 60 s (sample 120), 90 s (sample 180), and 120 s (sample 240) and shows the estimated temperature field across the 5 × 5 mesh as reconstructed by the embedded system solver. The color scale is fixed between 20 °C and 40 °C for consistency, while each cell displays its corresponding temperature in degrees Celsius. Additionally, the real sensor reading at the physical center of the plate is displayed below each frame, along with the computed estimate $\left(T_{13}\right)$ and the instantaneous estimation error.

The sequence reveals a gradual and symmetric thermal propagation from the edges toward the center of the plate. At 30 s, the boundary nodes are already responding to the applied heat, while the central region remains cooler. As time progresses, the isothermal curves become more centralized and rounded, indicating a diffusive propagation pattern consistent with two-dimensional conduction. By the 150 s mark, the internal temperature field is approaching a quasi-stationary state, with reduced thermal gradients between adjacent nodes.

The comparison between the estimated central value and the physical sensor confirms that, during this interval, the embedded digital twin provides a reasonably accurate approximation of the true thermal state, with deviations generally below ±3 °C. These visualizations validate the ability of the system to reconstruct the internal thermal profile from boundary-only measurements, highlighting the relevance of this approach for real-time diagnostics and heat distribution monitoring in embedded systems.

#### 5.5.2. Asymmetrical Perturbations Experiment

To investigate the response of the thermal estimation system under asymmetric excitation conditions, an experimental run was conducted in which only a single heat source was applied to one edge of the metallic plate. This setup was designed to evaluate how the internal estimation model, based on the Laplace equation, reacts to directional and non-uniform heat propagation.

In this experiment, the thermal evolution at the center of the plate was analyzed by tracking both the estimated temperature value computed from the boundary conditions (Figure 9a) and the physically measured value acquired through an embedded thermistor located at the geometric center of the plate. Both values were monitored continuously over a ten-minute interval, with uniform sampling assumed across the duration. Two snapshots were taken from the digital twin output to visualize the thermal performance, which are shown in Figure 9c,d.

The comparative analysis revealed that, while the Laplacian model provided a reasonable early-stage approximation, it increasingly underestimated the central temperature as the experiment progressed. This discrepancy is attributable to the inability of the stationary Laplace model to capture the dynamic accumulation of heat in the central region, which is influenced by the sustained unidirectional energy input from the single heat source.

Additionally, the thermal propagation from the edge to the center was quantified by examining the temporal evolution of a sequence of internal nodes aligned along the heat flow path, from the heated edge $T_{23}$ through intermediate positions $T_{18}$, to the center $T_{13}$ (Figure 9b). The resulting temperature profiles confirmed a delayed and diffused response consistent with one-dimensional heat conduction through a conductive medium, validating the system’s ability to visualize directional thermal gradients.

This asymmetric heating scenario demonstrates the utility of the embedded estimator not only for real-time visualization but also as a diagnostic tool to study spatiotemporal thermal behavior under non-ideal conditions, offering insight into the limitations of steady-state models when exposed to dynamic or localized stimuli.

Finally, to further improve the interpretability of the embedded estimator, Figure 10 shows a comparative vector field that includes both the temperature gradient and the resulting heat flow. In subplot (a), the temperature gradient vectors point toward regions of higher temperature, showing how the scalar field changes under asymmetric boundary excitation. In contrast, subplot (b) displays the heat flow vectors based on the negative gradient, illustrating the actual direction of thermal energy movement from hot to cold areas, as explained by Fourier’s law. This dual visualization offers directional insight that scalar heatmaps alone cannot provide, emphasizing how localized boundary conditions distort the thermal field and break radial symmetry. Notably, the consistent directional pattern around the central sensor $T_{13}$ acts as an intuitive confirmation of the spatial consistency of the embedded estimator.

### 5.6. Performance Comparison

To contextualize the capabilities of the proposed implementation in this work, a qualitative and quantitative comparison is provided against two state-of-the-art approaches: a digital accelerator for PDE solving [12] and an analog CMOS-based solver [13]. These references exemplify distinct design philosophies, with [12] emphasizing high-speed digital parallelism and [13] showcasing continuous-time analog computation. The comparisons shown in Table 1 and Table 2 outline key architectural, functional, and performance-related differences, demonstrating how the proposed solution achieves a balanced trade-off between accuracy, power efficiency, scalability, and system integration, particularly suited for embedded, real-time field reconstruction applications.

As can be seen in Table 1, the proposed embedded implementation offers a compact and physically interpretable solution for real-time thermal field reconstruction, standing in contrast to the more specialized and resource-intensive approaches in [12,13]. While [12] employs a high-performance SRAM-based process-in-memory accelerator to solve the 2D Poisson equation using multigrid residual methods, it requires custom silicon and is optimized for desktop-class scientific computing. In contrast, Ref. [13] realizes a nonlinear shock tube PDE solver using analog CMOS hardware, achieving high analog bandwidth but at the cost of high-power consumption (936 mW) and limited programmability. The proposed work solves the 2D Laplace equation using a physics-informed digital twin implemented on a low-cost STM32 microcontroller, integrating real-time sensing, computing, and communication under 100 mW of power. Unlike [12,13], which are fixed to their respective problem classes and hardware substrates, the proposed solution is fully programmable, modular, and adaptable to a range of physical systems governed by elliptic PDEs. The design enables physically consistent monitoring without the need for external computing resources or analog calibration, making it highly suitable for educational, prototyping, and diagnostic applications at the edge.

From Table 2, the proposed implementation demonstrates a favorable balance between performance, power consumption, and hardware footprint compared to references [12] and [13], particularly when considered in the context of embedded deployment. Reference [12] achieves a remarkable 1.38 G grid updates per second with a multigrid PDE solver built on a custom 1.87 mm^2^ 180 nm MAC-SRAM accelerator, consuming 16.6 mW at 200 MHz. Reference [13], while operating in continuous time with an 80 MHz equivalent update rate, incurs significantly higher power consumption (936 mW) and occupies a much larger silicon area (34.24 mm^2^), reflecting the analog complexity required for nonlinear PDEs. In contrast, the proposed system achieves approximately 500 updates per second on a configurable 5 × 5 grid using an off-the-shelf STM32 microcontroller. Though the absolute update rate is lower, the proposed work consumes under 100 mW, occupies less than 0.05 mm^2^ of silicon (MCU core area), and eliminates the need for external hardware or complex calibration. In terms of accuracy, [12] reaches a numerical error tolerance of 10^−8^, and [13] achieves a normalized MSD of −11.5 dB after calibration. The proposed implementation yields a relative error around 5%, which is acceptable for real-time field estimation in constrained environments. Therefore, although not targeting the same absolute computational throughput or precision, the presented solution outperforms in terms of system integration, portability, and energy efficiency, enabling practical and scalable embedded PDE modeling with minimal overhead.

## 6. Discussion

The implementation of a physics-informed digital twin for estimating real-time thermal distribution in metallic structures demonstrates both the feasibility and educational value of the proposed approach using mainstream microcontrollers. The embedded system can reconstruct internal thermal states by utilizing the steady-state Laplace equation and Dirichlet boundary conditions obtained from calibrated thermistors, achieving high temporal resolution with low computational overhead.

One of the advantages of this approach lies in its reliance on first-principles modeling, which ensures the physical consistency and interpretability of results without needing extensive training datasets typical of purely data-driven methods. The Laplace solver employs the Jacobi method with over-relaxation, achieving stable solutions in a limited number of iterations, making it suitable for real-time applications. While the Jacobi method was chosen for its simplicity and low memory use, it is known that more advanced solvers, like the Conjugate Gradient or Generalized Minimal Residual, could achieve faster convergence for large grids. However, these methods involve global operations like vector dot products, and they require extra memory to store search directions or Krylov subspaces, which may not be feasible on low-power microcontrollers. Future work may explore customized versions of these solvers or preconditioned schemes optimized for embedded platforms with hardware floating-point support.

The visualization layer built in Python allows users to see results similarly to how they would through a thermal camera, turning a physical setup into an interactive, real-time platform. This pairing of actual hardware with its simulated version creates a digital twin, effectively linking theoretical ideas with practical application.

However, several challenges can be pointed out. First, the spatial resolution is inherently constrained by the number of boundary sensors and the grid size that can be implemented within the available SRAM memory. Although a 5 × 5 grid proved sufficient for demonstrating thermal gradients, finer discretization would require optimization in memory allocation or the use of external RAM. Additionally, using floating-point arithmetic on microcontrollers can lead to precision and performance issues in certain setups. This work used a microcontroller with built-in hardware support for floating-point operations, which avoided these limitations. However, other platforms may not offer this support, and in such cases, convergence and accuracy may suffer, particularly when dealing with sharp thermal gradients or meeting industrial accuracy standards.

Sensor calibration with the Steinhart–Hart model was performed effectively in this laboratory setup to maintain accuracy. However, thermistor nonlinearity and potential thermal lag at the boundaries may result in minor deviations in the reconstructed data values. Operating directly at the edge computing level, where sensing, computation, and decision-making occur on the embedded platform itself, minimizes latency and reduces data transmission requirements. However, this architecture also imposes strict constraints on synchronization and timing. While utilizing DMA significantly reduces CPU overhead during ADC data acquisition, it presents timing challenges that must be addressed to maintain coherence between measurement and computation. In this implementation, interrupt-driven synchronization routines were required to precisely manage the capture and update cycles of both DMA and ADC peripherals, ensuring reliable real-time performance within the embedded environment.

Although the embedded solver relies on the steady-state Laplace equation, the digital twin maintains state concurrency by constantly updating boundary conditions in real time using calibrated thermistors. The system responds dynamically to boundary changes, allowing internal field reconstructions to track the physical evolution of the system. This real-time updating of internal states guarantees functional concurrency between the physical and digital layers, a fundamental characteristic of digital twins. Although transient behavior is not explicitly modeled, the system provides a quasi-static approximation suitable for situations where thermal dynamics change slowly compared to the update frequency.

From a systems perspective, the modular design of the embedded digital twin, which encompasses acquisition, computation, communication, and visualization components, presents opportunities for future enhancements. The anticipated improvements may involve real-time actuation (e.g., cooling systems), cloud-based logging, or hybrid physics-informed/machine learning models that adapt parameters over time.

The temperature reconstruction within this work aligns mathematically with traditional inverse heat transfer problems. However, unlike traditional formulations that are often solved offline and in isolation, the proposed system operates in real time, with continuous boundary collection and instant state estimation. This real-time synchronization, or state concurrency, sets digital twins apart from inverse models. Moreover, the broader architecture includes sensing, computation, and communication layers that enable operational feedback, making the system adaptive and extensible beyond conventional inverse approaches [5].

Although the experiment was conducted in a laboratory without thermal isolation, all system parameters, such as sensor calibration, boundary excitation, and acquisition timing, were meticulously controlled. The purpose of this setup was to evaluate the robustness of the digital twin under real-world ambient fluctuations. However, to measure the model’s true accuracy, future studies will test it under thermally shielded conditions to determine the system’s baseline performance without external disturbances. The consistent spatiotemporal patterns observed across trials indicate that environmental artifacts do not cause the results but are due to the physics-informed numerical reconstruction.

It is important to note that the number of internal grid points is not fixed; the resolution of the thermal field can be increased by adding more sensors along the boundary. The solver architecture remains unchanged, making the system scalable in spatial resolution as long as sensing density and memory resources are adjusted proportionally.

From the practitioner’s point of view, the embedded digital twin framework presented in this study validates the practicality of real-time simulation of partial differential equations on resource-constrained platforms. Its affordability, ability to replicate, and notable experimental value make it a valuable resource for industrial diagnostics and engineering education.

## 7. Conclusions and Future Work

This work presented a physics-informed digital twin architecture capable of estimating and visualizing the resulting thermal pattern under steady-state conditions in real-time. The system integrates boundary temperature measurements obtained through experimentally calibrated thermistors, a numerical solver based on the Laplace equation using the finite difference method, and an embedded implementation on a low-power mainstream microcontroller using DMA-driven ADC acquisition. The computed thermal map is transmitted and visualized through a Python interface, allowing an intuitive interpretation of the results.

The findings showed that real-time physics-based modeling can be effectively implemented on resource-limited embedded systems while maintaining accuracy and responsiveness. The Jacobi method’s use of overrelaxation enabled rapid convergence while maintaining simplicity of implementation. This approach is modular, extensible, and low-cost, making it ideal for educational settings and preliminary prototyping of thermal systems.

The current model is valid under quasi-static conditions and is not intended to replace full transient thermal models. Its purpose is to provide real-time estimation of spatial fields in systems where steady-state assumptions are applicable within short time windows.

An interesting extension and future work would be implementing a transient heat conduction model to more accurately capture the system’s dynamic response during the initial heating stages. Although more precise, such models require more memory and computing power to store time-dependent states and run time-marching schemes, which could exceed the capacity of lower-end microcontrollers.

Several future directions are anticipated. Firstly, increasing spatial resolution through adaptive meshing or memory optimization techniques would enhance model fidelity. Secondly, expanding the framework to manage dynamic thermal behavior would enable modeling of heat propagation and thermal inertia. Thirdly, integrating actuation mechanisms based on feedback control from the reconstructed thermal field would close the loop and facilitate autonomous thermal regulation. Finally, hybrid approaches that combine physical modeling with lightweight machine learning techniques could improve robustness against sensor noise and parameter uncertainty.

## Figures and Tables

**Figure 1 sensors-25-05130-f001:**
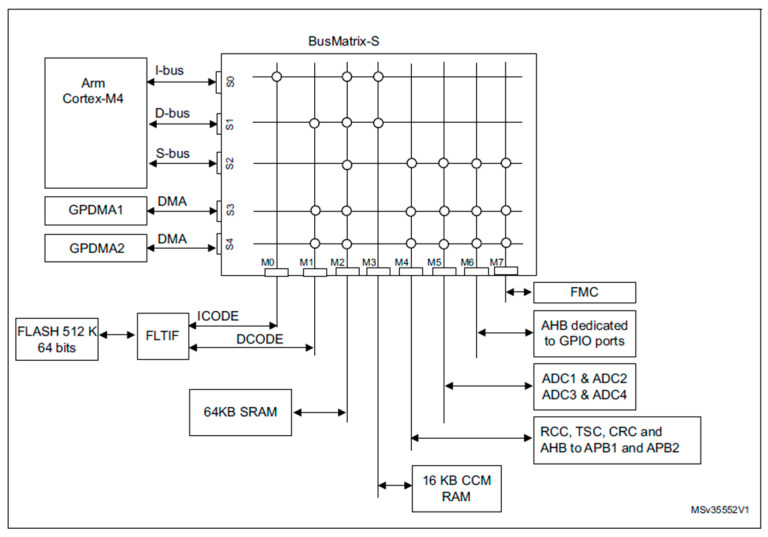
BusMatrix architecture depicting the DMA component and its integration to transfer peripheral data to memory in the ARM Cortex-M4 embedded systems.

**Figure 2 sensors-25-05130-f002:**
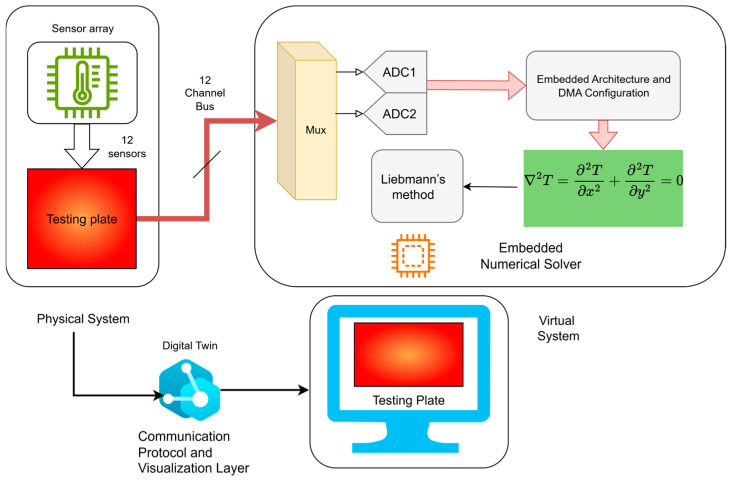
Components of the proposed methodology.

**Figure 3 sensors-25-05130-f003:**
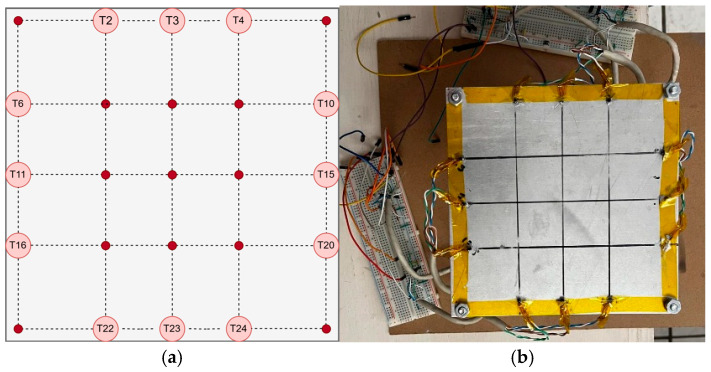
(**a**) Schematic diagram of thermal field reconstruction using a 5 × 5 mesh: red dots represent estimated interior temperatures, and circled values correspond to boundary temperatures measured by thermistors; (**b**) Physical setup displaying the actual sensor deployment on the metallic plate.

**Figure 4 sensors-25-05130-f004:**
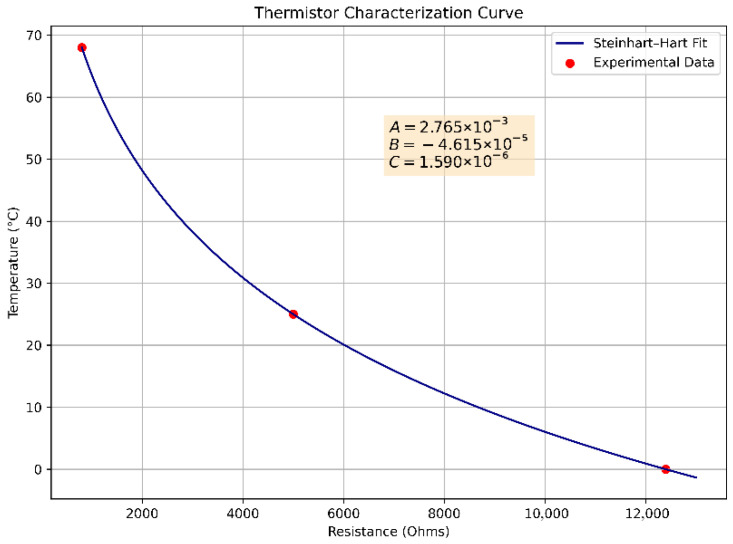
Experimental characterization curve of one thermistor. Each calibration point averages ten measurements, with a standard deviation of less than ±0.2 °C.

**Figure 5 sensors-25-05130-f005:**
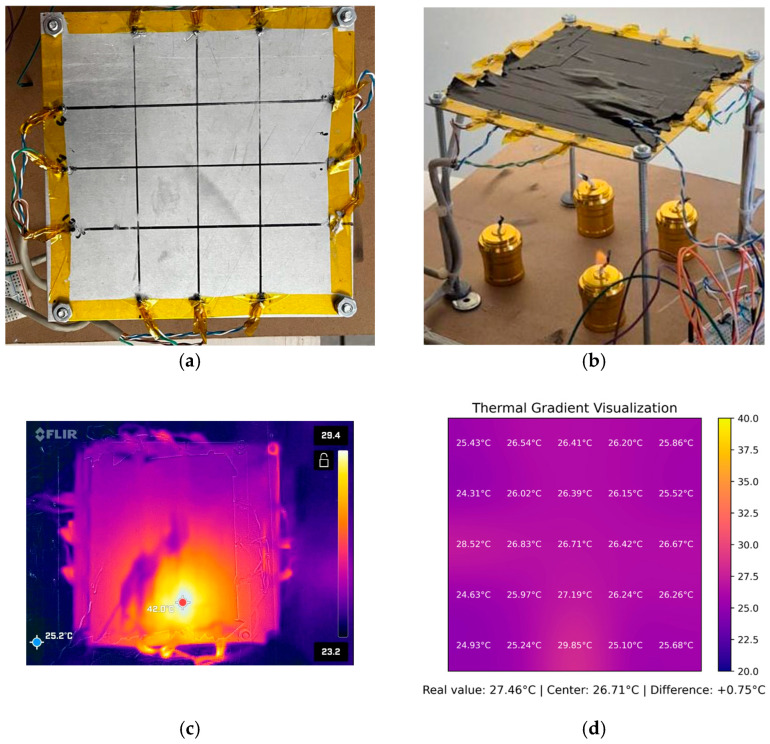
Experimental setup used during real-time tests; (**a**) close-up view of the sensor array; (**b**) external heat sources applied to the plate; (**c**) thermal images captured by a camera to assess the temperature gradient as an initial estimate, with a control sensor located at the center of the plate; and (**d**) gradient image computed by the digital twin.

**Figure 6 sensors-25-05130-f006:**
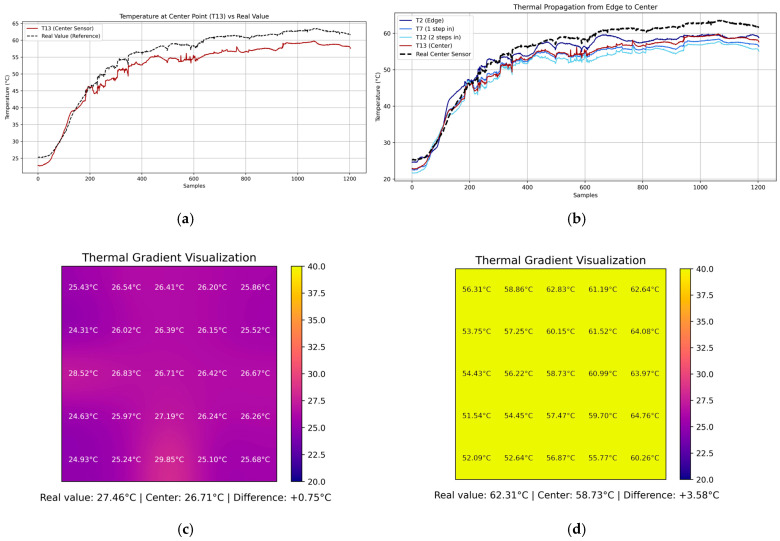
Experimental results under symmetrical heating conditions. (**a**) Temporal evolution of the estimated center temperature $\left(T_{13}\right)$ compared to the measured reference sensor. (**b**) Thermal propagation along a linear path from the top edge $\left(T_{2}\right)$ to the center $\left(T_{13}\right)$, showing delay and diffusion consistent with Fourier’s law. (**c**) The reconstructed thermal field at 31.5 s shows early-stage uniform distribution. (**d**) The final-stage thermal field at 10 min shows near-equilibrium conditions.

**Figure 7 sensors-25-05130-f007:**
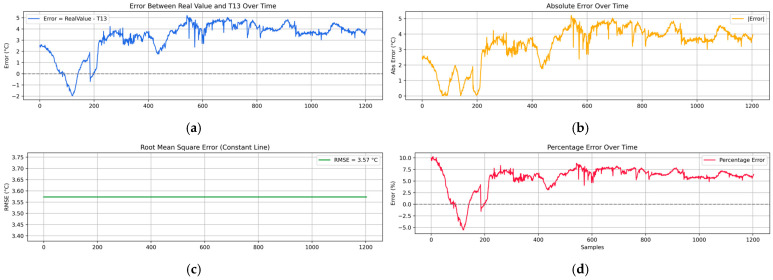
Error metrics between the estimated center temperature $\left(T_{13}\right)$ computed by the embedded digital twin and the physically measured reference sensor during the symmetrical heating experiment. (**a**) Time-resolved error between the reconstructed and real temperature at the center. (**b**) Absolute error magnitude over time. (**c**) Constant Root Mean Square Error (RMSE) line across the experiment duration (RMSE = 3.57 °C). (**d**) Percentage error evolution relative to the ground-truth measurement.

**Figure 8 sensors-25-05130-f008:**
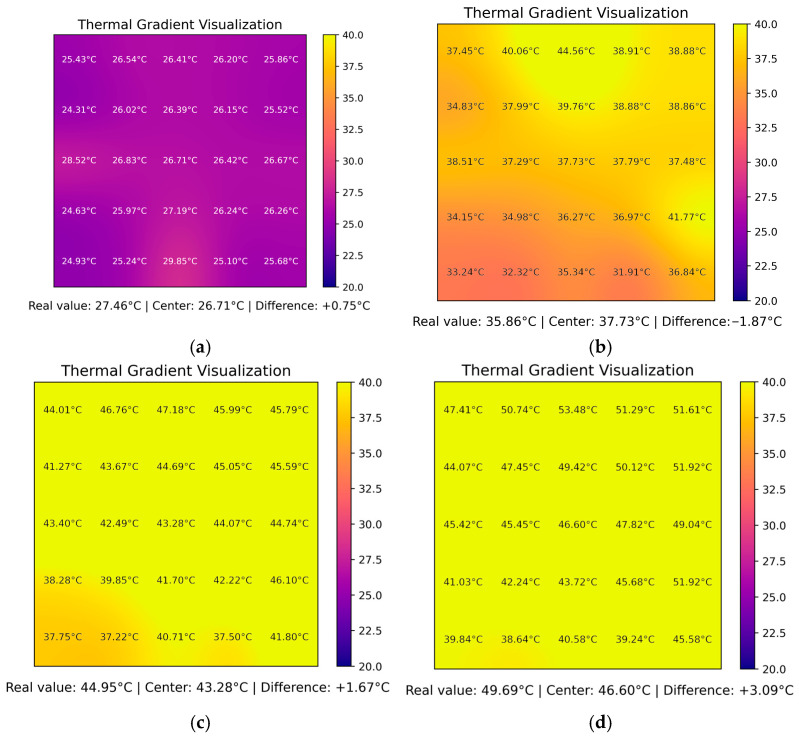
Real-time heatmap snapshots from the digital twin interface during the first three minutes of the symmetrical heating experiment. Each subfigure corresponds to a distinct time point: (**a**) 30 s, (**b**) 60 s, (**c**) 90 s, and (**d**) 120 s.

**Figure 9 sensors-25-05130-f009:**
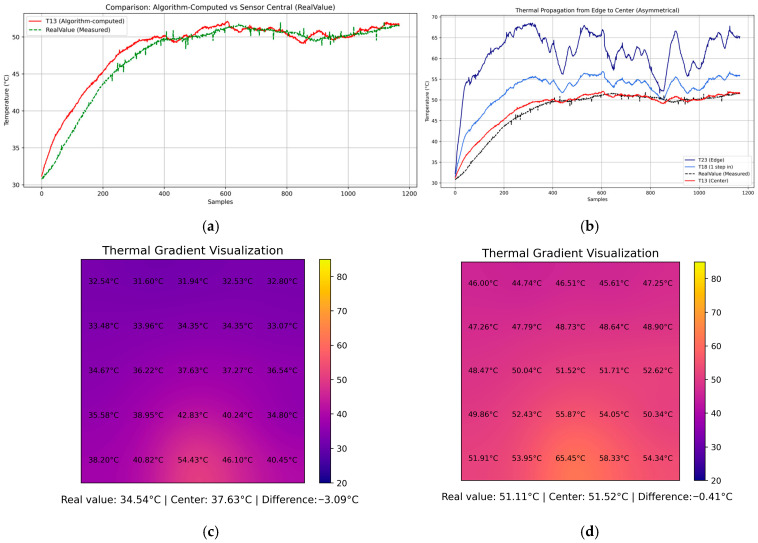
Experimental results under asymmetrical heating conditions. (**a**) Temporal evolution of the estimated center temperature $\left(T_{13}\right)$ versus the actual sensor measurement during a 10 min test, where a single heat source was applied to one edge of the metallic plate. (**b**) Propagation path from the heated edge to the center: the temperature evolution along nodes $T_{23}$ → $T_{18}$ → $T_{13}$ reflects directional heat diffusion and delayed response over time compared to the actual value. (**c**) Snapshot of the reconstructed thermal field during the early stage (≈30 s), showing minimal thermal spread; estimated center temperature: 37.63 °C versus real value: 34.63 °C. (**d**) Snapshot near the final stage (≈10 min), showing directional thermal convergence with a residual error of −0.41 °C.

**Figure 10 sensors-25-05130-f010:**
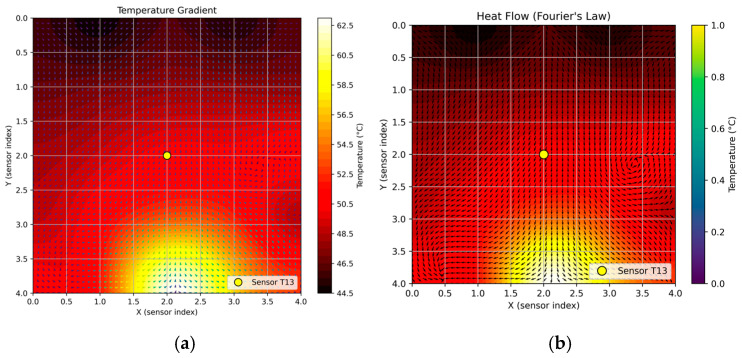
Vector field representation under asymmetric edge heating. (**a**) Temperature gradient vectors point toward regions of higher temperature. (**b**) Heat flow vectors obtained from the negative temperature gradient, illustrating energy propagation from hot to cold zones. The central sensor *T*_13_ is highlighted in yellow in both plots.

**Table 1 sensors-25-05130-t001:** Qualitative comparison.

Feature	[12] PDE Accelerator	[13] Analog Solver	Our Work
PDE Type	2D Poisson	Nonlinear Acoustic Shock Tube PDE	2D Steady-State Laplace
Computation Platform	MAC SRAM PIM (Digital)	Analog CMOS (TSMC 180 nm)	STM32 MCU (Embedded)
Precision Strategy	5-bit residual multigrid	Analog delay line, calibration	Float/Fixed-point, iterative
Real-Time Capability	Yes	Yes	Yes
Scalability	Moderate (128 × 128 grid)	High (15 spatial points)	High (grid configurable)
Use Case	Scientific Desktop Apps	Real-Time Physical Modeling	Thermal Diagnostics/Prototyping
Programmability	Medium (Multigrid fixed)	Low (fixed architecture)	High (Embedded FDM solver)
Power Efficiency Focus	High	Very High	Moderate to High

**Table 2 sensors-25-05130-t002:** Quantitative comparison.

Metric	[12] PDE Accelerator	[13] Analog Solver	Our Work
Update Rate/Grid Speed	1.38 G updates/s	80 MHz equivalent	~500 updates/s (5 × 5 grid)
Power Consumption	16.6 mW @200 MHz	936 mW	<100 mW (STM32)
Chip/Platform Area	1.87 mm^2^	34.24 mm^2^	~0.05 mm^2^ (MCU core)
Normalized MSD (Accuracy)	10^−8^, tolerance	−11.5 dB	Relative error ~5%
Grid Size	128 × 128	15 spatial points	5 × 5 (configurable)

## Data Availability

The data supporting the findings of this study are available from the corresponding author upon reasonable request. Due to the nature of the experimental setup and institutional policies, the dataset is not publicly archived but can be shared for research and academic purposes upon direct communication with the authors.

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
