# Peer review of "Thermal Field Reconstruction on Microcontrollers: A Physics-Informed Digital Twin Using Laplace Equation and Real-Time Sensor Data"

_sensors, 2025, doi:10.3390/s25165130_

Round 1

Reviewer 1 Report

Comments and Suggestions for Authors

This study presents a physics-informed digital twin for solving the Laplace heat conduction equation on an embedded platform. The implementation employs DMA (Direct Memory Access) and ADC (Analog-to-Digital Converter) for real-time sensor data acquisition using the STM32 microcontroller series. The governing equation is discretized using the finite difference method (FDM) and solved iteratively via the Jacobi method. Boundary conditions are dynamically updated using pre-calibrated thermistors, ensuring accurate thermal modeling. The embedded system optimizes computational efficiency while preserving physical consistency with the real-world thermal process. While the manuscript is well-structured and technically sound, the following aspects could be further improved:

1 – Typos: In line 195, on the title; choose equation or Equation in line 268; The title in line 301 and 309 are the same; Figure 13 is not present, in line 432.

2 – The timestep should be explicitly indicated for each subfigure in Figure 8 to aid interpretation.

3 – Inverse problems in heat transfer estimate internal temperature distributions from boundary measurements. Since this work also uses boundary data (acquired via thermistors) to model thermal behavior, a discussion on the key distinctions between a digital twin and an inverse problem application would be valuable for readers.

4 – The authors stated errors in the manuscript due to transient effects in estimating internal plate temperatures. Could a transient heat conduction model improve sensor response? If so, what would be the additional memory requirements for such an implementation?

5 – The Jacobi method, while simple, has slow convergence. Would more advanced methods like the Conjugate Gradient or Generalized Minimal Residual Method (GMRES) be feasible for real-time processing? An analysis of their potential benefits and computational trade-offs would strengthen the discussion.

6 – What is the time processing between each thermal image?

Reviewer 2 Report

Comments and Suggestions for Authors

The paper claims to have developed a physics-informed digital twin. However, the definition of DT is not rigid and the paper does not identify attributes that must be met to be defined as a DT. What makes the model developed in this paper a DT when compared to existing definitions of DT?

3.1 Laplac's -> spelling

Eqn. 7 Use multiplication symbol

Figure 2 Digital Twink -> spelling

The spacing of the grid matrix (shown in line 364) for predicting temperature points seems to be a course grid. The justification was to balance computational load. Describe the spacing of the grid and the impact on computational requirement for different spacing considerations. 

Line 428 should say figure (c) and (d)

Assuming that the orientation of Figure (c) and (d) are the same, the experimental results do not support the thermal gradient visualization. In (c), the northern edge is cool while in the (d) the northern edge is clearly much hotter, by about 10C. Furthermore, the hottest point (42C) near the southern edge is opposite to the visualization. Discuss these differences in results. 

A property of a digital twin is its ability for state concurrency. Line 518 and 519 state that the formulation assumes steady-state heat conduction and neglects transient behavior. If this is the case, then the argument that the application is a DT is weak. 

The justification provided in 527 about a controller laboratory environment is odd. The purpose of a controlled environment is to know the exact capabilities of the system. Testing in an uncontrolled environment first without determining its capabilities in a controlled environment introduces too many uncertainties in the conclusion. How do we know that the results are not just a fluke due to environment conditions? 

Better figure labels are required. Especially for Figure 6 and 8 which have visualizations but not description of what is observed. Also, change the temperature values to black rather than white, which is currently invisible with a yellow background.

9c and 9d are snapshots are different intervals. What are the exact intervals?

The highlights of this paper claims to have developed a compact, scalable, and physically consistent thermal diagnostics tool. However, the discussions points to several difficulties that contradict this highlight. Grid resolution limits implies scalability limitations; the need for external ram implies not compact; and (while not significant) the error in transient response suggest that the current approach is not entirely physically consistent with experimental results. In essence, the novelty of this work is not clear.

A larger concern is the claim of a digital twin. The paper does not make any explicit effort into justifying why the model developed is considered a digital twin; does not have any of the features (state concurrency, state cognizance, data storage/retrieval/processing, etc.) that other papers have claimed are required for a digital twin. 

Reviewer 3 Report

Comments and Suggestions for Authors

Comments-sensors-3774647
The article seems interesting to me. In this work, the author presented a physics-informed digital twin using Laplace equation and real-time sensor data. The manuscript looks technically sound to me; however, it needs to be revised to address my following comments:
1.    I am concerned with their ‘Results and Discussion,” where all the figures need elaborate explanations.
2.    In Figure 4, the authors plotted the graph based on only 3 data points. Three data points are not statistically reliable. It is advised to have at least 5 data points to avoid the impact of the outliers.
3.    In Figure 4, what is the frequency of each data point? It’s advised to plot with the error bars and mention the corresponding SD values in the manuscript.
4.    In line 428, the authors mentioned “Figure 4 (b) and (c) show …..”It should be 5 (b) and (c).
5.    In conclusion, the major quantitative results should be included. 

Reviewer 4 Report

Comments and Suggestions for Authors

The manuscript presents a physics-informed digital twin system implemented on embedded hardware to reconstruct steady-state thermal fields in real time. The work is original, well-structured, and shows promise for applications in edge computing and thermal diagnostics. However, I suggest the following improvements to enhance clarity and scientific rigor:

1. While the related work section provides a broad overview of existing digital twin frameworks, the manuscript lacks a detailed comparison between the proposed system and prior PDE-based embedded solutions. Including benchmark metrics or performance comparisons (e.g., computation speed, power efficiency, or accuracy) with works such as [13] or [14] would better illustrate the advantages of this implementation.

2. The current results section demonstrates thermal field visualization, but there is no clear error assessment between reconstructed and ground-truth values. Adding numerical metrics such as RMSE, maximum deviation, or percentage error across the grid would significantly strengthen the validation and credibility of the results.

3. In the text referring to Figure 3, the distinction between Figure 3a and Figure 3b is unclear. Please revise the captions and in-text references to explicitly indicate which subfigure represents the schematic and which shows the physical implementation.

4. Sections 4.3 and 4.4 share the title “Communication and Data Exchange,” with overlapping content. Consider merging these sections or refining the distinctions in scope. Additionally, some expressions (e.g., “These readings serve as Dirichlet boundary conditions”) are repeated multiple times and could be phrased more concisely for readability.

With the above revisions, I believe the manuscript will be well-suited for publication in *Sensors*. 

Round 2

Reviewer 1 Report

Comments and Suggestions for Authors

I thank the authors for the enhanced version of the manuscript. This reviewer believes the manuscript is ready to be published.

Reviewer 3 Report

Comments and Suggestions for Authors

The authors addressed all the comments satisfactorily. I have no more concerns on the technical parts. However, the english can still improve.